# Couple’s Relationship during the Transition to Parenthood and Toddler’s Emotional and Behavioral Problems

**DOI:** 10.3390/ijerph20010882

**Published:** 2023-01-03

**Authors:** Tiago Miguel Pinto, M. Carmen Míguez, Bárbara Figueiredo

**Affiliations:** 1School of Psychology, University of Minho, 4710-057 Braga, Portugal; 2HEI-Lab: Digital Human-Environment Interaction Lab, Lusófona University, 1749-024 Lisboa, Portugal; 3Department of Clinical Psychology and Psychobiology, Faculty of Psychology, University of Santiago de Compostela, 15705 Santiago de Compostela, Spain

**Keywords:** couple’s relationship, toddlers, emotional and behavioral problems, internalizing and externalizing problems

## Abstract

The literature has mainly focused on the impact of the negative aspects of the couple’s relationship on the toddler’s internalizing and externalizing problems. This study explored the impact of the positive and negative dimensions of the couple’s relationship during the transition to parenthood on toddlers’ psychological adjustment, considering the concurrent impact of the couple’s relationship at 30 months postpartum. The sample comprised 115 mothers and fathers (*N* = 230) recruited during the 1st trimester of pregnancy. The mothers and fathers individually completed a measure of the couple’s relationship (Relationship Questionnaire) during the first trimester of pregnancy, at 3 and 30 months postpartum, and the Child Behavior Checklist 1.5–5 at 30 months postpartum. Multiple linear regressions, cluster analyses, and univariate and multivariate analyses of variance were conducted. The positive dimension at the 1st trimester of pregnancy and the negative dimension of the couple’s relationship at 3 months postpartum were the strongest predictors of the toddler’s internalizing problems, while the negative dimension at 3 months postpartum and the positive dimension of the couple’s relationship at 30 months postpartum were the strongest predictors of the toddler’s externalizing problems. Two patterns of the couple’s relationship (adjusted vs. non-adjusted) during the transition to parenthood were identified. Higher levels of internalizing and externalizing problems were found in toddlers from couples with a non-adjusted couple’s relationship. Findings suggested the impact of both positive and negative dimensions of the couple’s relationship during the transition to parenthood on the toddler’s emotional and behavioral problems. Promoting the couple’s relationship adjustment during the transition to parenthood can help to prevent toddlers’ emotional and behavioral problems.

## 1. Introduction

Becoming a parent is a normative transition [1]. The transition to parenthood is defined as a long-term process that results in a qualitative reorganization that allows fulfilling, successfully or not, the parenting-related tasks [1]. Although it is not undisputed that the couple’s relationship worsens during the transition to parenthood, changes occur in several aspects of the couple’s relationship, including division of tasks, communication, intimacy, satisfaction, support, closeness, negative affect, anxiety, or irritability [2,3,4]. The dyad can cope with this developmental transition in different ways, maintaining an adjusted or non-adjusted couple relationship balance [5]. Although there is variability in the functioning of the couple’s relationship associated with the demands and the experiences of the transition to parenthood [6,7,8], studies do not focus on features of the couple’s transition to parenthood in a more or less adaptive way. Literature reports that the couple’s relationship can decline from early in pregnancy [9,10,11,12,13]. The intimacy and marital satisfaction tend to decrease, while conflicts and the disconfirmation of expectancies tend to increase from pregnancy to the postpartum period [14].

Nevertheless, research lacks consensus on the decline in the couple’s relationship while in the transition to parenthood [14]. Meta-analytic research has found small decreases in couple satisfaction from pregnancy to the postpartum period, which is similar to what was found in childless couples [15], suggesting that the observed relational decline was not a result of the transition to parenthood but was frequent during the first years of marriage. Other studies on marital satisfaction in couples with or without children were also meta-analytically reviewed and higher levels of marital satisfaction were reported in childless’ couples [16]. Empirical evidence of the non-adjusted versus adjusted profiles of the couple’s relationship during the transition to parenthood can contribute to explaining the mixed findings regarding the decline of the couple’s relationship from pregnancy to the postpartum period [14,15,16]. However, empirical evidence on the profiles of the couple’s relationship during the transition to parenthood is needed.

According to the spillover model, the couple’s relationship during the transition to parenthood can impact both parenting and co-parenting [17,18], parents’ psychopathology [19,20], and the security within the triad [21,22], which can impact child development. The couple’s relationship quality can impact the quality of the care that is provided to the child [23]. Higher levels of marital conflict have been consistently associated with children’s externalizing and internalizing problems, in several studies and different contexts [24,25,26,27]. The impact of the couple’s negative relationship quality on the child’s psychological adjustment has been mostly studied through the examination of the negative dimension of the couple’s relationship [24,25,26,27]. However, the emotional, behavioral, and social positive adjustment during toddlerhood seems to be predicted by the positive dimension of the couple’s relationship during the transition to parenthood [4,28]. Studies reported that the positive aspects of the couple’s relationship, such as expressing love or enjoying being together, can have a protective impact against the emergence of the toddler’s internalizing and externalizing problems [23,29]. 

Parental gender can also play a role in the impact of the couple’s relationship on the toddler’s psychological adjustment. Father’s perspective on the marital relationship was found to better predict the child’s externalizing problems, than the mother’s perspective [30]. Still, other studies described that when the mother reports a sharper decline in the couple’s satisfaction, children’s internalizing problems tend to occur at a higher rate, while the father’s satisfaction with the couple’s relationship does not impact the children’s internalizing problems [31]. 

Although an association between the couple’s relationship quality during the transition to parenthood and children’s externalizing and internalizing problems has been reported, studies have been focused mainly on the negative aspects of the couple’s relationship (e.g., conflicts) [24,25,26,27], while the positive aspects (e.g., joint activities) may also be important to understand the process underlying the impact couple’s relationship on children’s psychological adjustment. Likewise, exploring the impact of a non-adjusted versus adjusted couple’s relationship across the transition to parenthood on children’s psychological adjustment may be important to understand the long-term impact of the couple’s relationship quality on children’s early development. This study aimed to explore the impact of the positive and negative dimensions of the couple’s relationship during the transition to parenthood on the toddler’s externalizing and internalizing problems, considering the impact of the concurrent couple’s relationship at 30 months postpartum. Specifically, this study aimed (1) to identify which specific periods (pregnancy or postpartum period) of the positive and negative dimensions of the couple’s relationship during the transition to parenthood are the stronger predictors of the toddler’s psychological adjustment; and (2) to analyze the impact of non-adjusted versus adjusted couple’s relationship trajectories during the transition to parenthood on toddlers’ psychological adjustment. 

## 2. Materials and Methods

### 2.1. Participants and Procedures 

The sample comprised 115 mothers and fathers (*N* = 230) from a larger longitudinal study. The total sample was 260 couples, recruited from a northern Portuguese Obstetrics Unit during the first pregnancy trimester. 

The present research followed the Helsinki Declaration and received previous approval from the Ethical Commission of all institutions involved. Participants were recruited at a public Health Service in Northern Portugal during the first trimester of pregnancy. The study exclusion criteria were not reading or writing Portuguese and twin gestations. The aims and the procedures of the study were explained to the mothers and fathers and those who were willing to participate provided a written consent form (*n* = 520, 86.7%). This study had a longitudinal design with three assessment waves: (1) first trimester of pregnancy (1st TP, 8–14 gestational weeks, *M* = 12.97, *SD* = 1.49), (2) 3 months postpartum (10–14 postpartum weeks, *M* = 13.64, *SD* = 0.81), and (3) 30 months postpartum (21–46 months postpartum, *M* = 32.22, *SD* = 6.47). Both mothers and fathers independently completed a socio-demographic questionnaire, a measure of the couple’s relationship at each assessment wave, and they completed the Children Behavior Checklist (CBCL) 1.5–5 at 30 months postpartum. 

### 2.2. Measures

#### 2.2.1. Socio-Demographic Measures

A self-administered questionnaire was completed by mothers and fathers to collect mothers’, fathers’, (e.g., nationality, age, socioeconomic level, education, occupational status, marital status, and parity), and toddlers’ (e.g., sex, age, health problems, and daycare) socio-demographic information.

#### 2.2.2. Couple’s Relationship Quality

The Relationship Questionnaire (RQ) [32] was administered to assess fathers’ and mothers’ positive and negative dimensions of the couple’s relationship. It is a brief questionnaire that contains 12 items scored on a 4-point Likert-type scale. The positive dimension assesses the sense of support and care, affection, closeness joint interests, and activities (8 items). The negative dimension evaluates anxiety, irritability, and criticism (4 items). The total score for each dimension is computed by averaging the scores of the items of each subscale. This questionnaire was designed to be completed in a short time, is behaviorally focused, and has been used to assess the couple’s relationship during the transition to parenthood [2,13]. The RQ has shown good internal consistency and validity. In the present sample, the RQ presented good Cronbach’s alphas values for both the mothers (positive scale ranging from 0.80 to 0.92, and the negative scale from 0.73 to 0.81) and the fathers (positive scale ranging from 0.82 to 0.89, and the negative scale from 0.70 to 0.78). 

#### 2.2.3. Emotional and Behavioral Problems

The Child Behavior Checklist 1.5–5 (CBCL) [33] was administered independently to mothers and fathers to assess the toddlers’ internalizing and externalizing problems. This instrument consists of 99 items that describe behavioral and emotional problems in children of 1.5 to 5 years of age. Parents are requested to rate their toddler’s behavior during the last 2 months on a 3-point Likert-type scale. The CBCL assesses the following behaviors: emotional reactivity, anxious/depressed, somatic complaints, withdrawn, sleep problems, attention problems, and aggressive behavior. Based on second-order factor analyses, the first four syndromes formed a grouping that was designated “internalizing behavior”, while the last two formed a grouping designated as “externalizing behavior”. The psychometric characteristics of the CBCL have been analyzed in several countries [33]. The Portuguese version of the CBCL has shown good internal consistency and validity [34]. In the present sample, Cronbach’s alpha coefficient ranged between 0.90 and 0.94 for mothers’ and fathers’ reports of toddlers’ internalizing and externalizing scores.

### 2.3. Data Analyses

Analyses were performed using the SPSS 20 version (IBM, Armonk, NY, USA). Each parent RQ’s score was transformed into a composite score (an average score of the mother and the father RQ scores at each assessment wave was computed). Following previous recommendations [30], this procedure was adopted to better capture the dyadic nature of the data. According to previous recommendations, the same procedures were adopted for CBCL scores [35]. Preliminary analyses were performed to identify potential covariates. Univariate analyses of variance (ANOVAs) and multivariate analyses of variance (MANOVAs) were performed to analyze the potential impact of parents’ and toddlers’ sociodemographic variables on the studied variables.

Aim 1. Two multiple linear regressions (stepwise method) were computed to identify which moment of the positive and negative dimensions of the couple’s relationship during the transition to parenthood is the strongest predictor of toddlers’ psychological adjustment. Models included the RQ’s positive and negative composite scores at the 1st trimester of pregnancy, 3 and 30 months postpartum as independent variables (IVs), and parents’ reports of the toddlers’ internalizing and externalizing scores at 30 months postpartum as dependent variables (DVs).

Aim 2. Cluster analysis, two repeated measures ANOVAs, and a MANOVA were performed to analyze the impact of non-adjusted versus adjusted couple’s relationship trajectories during the transition to parenthood on toddlers’ psychological adjustment. Cluster analysis was performed (using the combinatory method) to discriminate between adjusted (ADJ) vs. non-adjusted (NADJ) couples’ relationship trajectories across the transition to parenthood. Parents’ positive and negative composite RQ scores at the three assessment waves were transformed into z scores and the centroids were calculated using the hierarchical procedure (Ward’s method and squared Euclidian distance). The centroids were used to compute the final clusters using the K-means method. The clusters were statistically validated using MANOVAs, including the clusters as IVs and the RQ’s positive and negative composite scores at each assessment wave as DVs. Additionally, two repeated measures ANOVAs were used to test for time and interaction effects between time and the clusters on the RQ’s positive and negative composite scores at each assessment. Another MANOVA was performed to test the impact of the couple’s ADJ vs. NADJ (IV’s) relationship on the toddlers’ internalizing and externalizing scores (DVs). Bonferroni corrections were applied for multiple comparisons.

## 3. Results

### 3.1. Participants’ Characteristics

Most of the parents were Portuguese (93.9% of mothers and 91.2% of fathers), the mother’s age was between 17 and 42 years of age (*M* = 29.81, *SD* = 5.59), and the father’s age was between 16 and 46 (*M* = 31.82, *SD* = 6.04). Most of the participants were from low or medium-low socioeconomic levels (74.5% of mothers and 66.4% of fathers), had nine or more years of education (80.9% of mothers and 67.8% of fathers), were employed (80.9% of mothers and 90.4% of fathers), and married or cohabiting (95.7%) (see Table 1). More than half were first-time parents (64.2%). The toddlers’ age at the third assessment wave was of 32 months of age (*M* = 32.22, *SD* = 6.47), the majority were male (56.5%) and were not attending preschool (50.4%). No toddler health problems were reported.

Of the 260 mothers and fathers (*N* = 520) that completed the first assessment wave, 128 mothers and 116 fathers (*n* = 244; 49.9%) completed the CBCL at 30 months postpartum. For this study, CBCL data were included only when completed by both parents (*N* = 230; *n* = 115 mothers; *n* = 115 fathers). Ninety-five couples (86.61%) responded to all assessment waves. The total sample (CBCL and the RQ scores) did not differ from those participants who did not complete the data on the three assessment waves.

### 3.2. Preliminary Analyses

No significant impact of any parental demographic variables was found on RQ’s parental scores (all *p*s > 0.05). Likewise, no significant impact of toddlers’ demographic variables was found on toddlers’ internalizing and externalizing scores (see Table 2). Therefore, none of these variables were included as a covariate in the following analysis.

### 3.3. Couple’s Positive and Negative Dimensions of the Couple’s Relationship at Each Assessment Wave as Predictors of the Toddler’s Internalizing and Externalizing Problems

#### 3.3.1. Internalizing Problems

The first predictor entering the regression equation was the negative dimension of the couple’s relationship at 3 months postpartum, *F*(1, 93) = 9.83, *p* = 0.002, explaining 9.6% of the variance. In the second step, both the negative dimension of the couple’s negative relationship at 3 months postpartum and the positive dimension of the couple’s relationship at the 1st trimester of pregnancy entered the equation, *F*(2, 92) = 7.99, *p* = 0.001. The model identified the negative dimension of the couple’s relationship at 3 months postpartum, *t*(92) = 2.44, *p* = 0.02, and the positive dimension of the couple’s relationship at the 1st trimester of pregnancy, *t*(92) = −2.38, *p* = 0.02, as the strongest predictors of toddlers’ internalizing problems. The negative dimension of the couple’s relationship at 3 months postpartum predicted higher levels of toddlers’ internalizing problems, whereas the positive dimension of the couple’s relationship at the 1st trimester of pregnancy predicted lower levels of toddlers’ internalizing problems. The model accounted for 15% of the variance, *R*^2^ = 0.15, *R*^2^_adj_ = 0.13.

#### 3.3.2. Externalizing Problems

The first predictor entering the multiple regression was the positive dimension of the couple’s relationship at 30 months postpartum, *F*(1, 93) = 17.05, *p* < 0.001, explaining 15.5% of the variance. In the second step, the positive dimension of the couple’s relationship at 30 months postpartum and the negative dimension of the couple’s relationship at 3 months postpartum entered both the regression model, *F*(2, 92) = 11.11, *p* < 0.001. The model identified the positive dimension of the couple’s relationship at 30 months postpartum, *t*(92) = −3.17, *p* = 0.004, and the negative dimension of the couple’s relationship at 3 months postpartum, *t*(92) = −2.13, *p* = 0.04, as the strongest predictors of toddlers’ externalizing problems. The positive dimension of the couple’s relationship at 30 months postpartum was associated with lower levels of toddlers’ externalizing problems, whereas the negative dimension of the couple’s relationship at 3 months postpartum predicted higher levels of toddlers’ externalizing problems. The model accounted for 20%, *R*^2^ = 0.20, *R*^2^_adj_ = 0.18, of the variance (see Table 3).

### 3.4. Impact of Adjusted vs. Non-Adjusted Couple Relationship Trajectories during the Transition to Parenthood on Toddler’s Internalizing and Externalizing Problems

#### 3.4.1. Clusters’ Analysis

The final solution integrated 47 couples in group 1 (49%; labeled adjusted couples) and 48 in group 2 (labeled non-adjusted couples). Convergence is achieved due to no or small changes in cluster centers. The minimum distance between initial centroids was 2.84.

#### 3.4.2. Clusters’ Validation

The multivariate tests, *F*(6, 88) = 41.67, *p* < 0.001, *ε*^2^_p_ = 0.74, the between-subjects effects tests and the pairwise comparisons were all significant (all *p*s < 0.001; see Table 4). At all assessment waves, the adjusted group reported higher scores on the positive dimension and lower scores on the negative dimension of the couple’s relationship than the non-adjusted group (see Table 4). Further repeated measures ANOVAs showed significant multivariate effects of the interaction between time and clusters on the positive and negative dimensions of the couple’s relationship (see Table 4). Pairwise comparisons applied to adjusted versus non-adjusted groups indicated that adjusted couples maintained high levels on the positive dimension of the couple’s relationship from the pregnancy to 3 months postpartum, and a decrease from 3 to 30 months postpartum. On the other hand, the adjusted group of couples presented low levels of the negative dimension of the couple’s relationship throughout the transition to parenthood, although with significantly higher levels at the 1st trimester of pregnancy than at 30 months postpartum. The non-adjusted couples presented high levels of the positive dimension of the couple’s relationship during the transition to parenthood, although from the 1st trimester of pregnancy to 30 months postpartum, the decline in the positive dimension of the couple’s relationship was significant from pregnancy to 30 months postpartum. Non-adjusted couples presented higher levels of the negative dimension of the couple’s relationship compared to the adjusted couples, maintaining constantly higher levels of the negative dimension of the couple’s relationship levels during the transition to parenthood, than the adjusted group (Table 4 and Figure 1). 

#### 3.4.3. Adjusted vs. Non-Adjusted Couple’s Relationship Impact on the Toddlers Internalizing and Externalizing Problems

The MANOVA’s multivariate results were significant, *F*(1, 93) = 5.61, *p* = 0.005, *ε*^2^_p_ = 0.11. The between-subjects tests were significant for toddler’s internalizing, *F*(1, 93) = 6.69, *p* = 0.01, *ε*^2^_p_ = 0.07, and externalizing problems, *F*(1, 93) = 11.30, *p* = 0.001, *ε*^2^_p_ = 0.11. Pairwise comparisons showed that toddlers from the non-adjusted couples presented significantly higher internalizing and externalizing problems when compared with toddlers from the adjusted couples (see Table 5). 

## 4. Discussion

This study explored the impact of the positive and negative dimensions of the couple’s relationship during the transition to parenthood on toddlers’ psychological adjustment, considering the concurrent impact of the couple’s relationship at 30 months postpartum. Findings provided evidence of the positive dimension of the couple’s relationship at the 1st trimester of pregnancy and the negative dimension of the couple’s relationship at 3 months postpartum as the strongest predictors of the toddler’s internalizing problems. On their turn, results suggested the negative dimension of the couple’s relationship at 3 months postpartum and the positive dimension of the couple’s relationship at 30 months postpartum as the strongest predictors of the toddler’s externalizing problems. Findings also revealed different couple’s relationship trajectories during the transition to parenthood for adjusted vs. non-adjusted couples. Adjusted couples revealed a decline in the negative dimension of the couple’s relationship from the antenatal to the postnatal period, while this decline was not found in non-adjusted couples. Additionally, both adjusted and non-adjusted couples presented a decrease in the positive dimension of the couple’s relationship from the 1st trimester of pregnancy to 3 and 30 months postpartum. However, this decrease was higher in non-adjusted couples. Finally, findings provided evidence on the impact of a non-adjusted versus adjusted couple’s relationship during the transition to parenthood on toddler’s internalizing and externalizing problems. Toddlers from the non-adjusted couples presented more internalizing and externalizing problems than toddlers from the adjusted couples.

In line with previous findings [36], the results of this study highlighted the importance of assessing longitudinally both the positive and negative dimensions of the couple’s relationship when analyzing its impact on toddlers’ emotional and behavioral psychological adjustment. Findings suggested that the positive aspects of the concurrent couple’s relationship can be protective against the toddler’s externalizing symptomatology at 30 months postpartum.

The study of the impact of the negative dimensions of the couple’s relationship on the toddler’s psychological adjustment has a larger tradition in the literature [24,25,26,27]. Similar to previous studies, the current study found evidence of the impact of the negative dimension of the couple’s relationship on toddlers’ psychological adjustment. The intensification of the negative dimension of the couple’s relationship from the pregnancy until the 30 months postpartum, and particularly the impact of the 3 months postpartum on the child’s psychological adjustment, can be enlightened by the demands of becoming a parent [9,16]. Our findings suggested that the moment of 3 months postpartum seems to be a critical point since only the negative dimension of the couple’s relationship at 3 months postpartum was identified as the stronger predictor of the toddler’s psychological adjustment. These findings are similar to those of previous studies, although not exactly during this period of the transition to parenthood [30,31]. Failure of parents to meet the prenatal expectations of marital conflict and cooperation might explain this impact [37]. The integration of prenatal expectations into the postnatal relationship is part of the complex process of adaptation to parenthood. When expectations are dashed, the decline in the quality of the couple’s relationship can be steeper. Thus, representing a risk factor for toddlers’ psychological adjustment. 

Another important finding of the present study concerns the identification of two different profiles of the couple’s relationship during the transition to parenthood. These results reinforce the evidence of the decline of the positive dimension of the couple’s relationship during the transition to parenthood for both adjusted and non-adjusted couples. However, adjusted couples were found to present a lower decrease in the positive dimension of the couple’s relationship than non-adjusted couples. Although this decline is in line with previous studies [9], the couple’s relationship trajectories found in adjusted and non-adjusted couples suggested that it is not only the relationship deterioration that can explain the negative impact on toddlers’ psychological adjustment problems. Instead, the negative impact could be explained by the extent of the decline, which can indicate the adaptation of the couple’s relationship to the transition to parenthood.

In addition, the present study found evidence that an adjusted couple’s relationship across the transition to parenthood could protect toddlers against the emergence of emotional and behavioral problems, while a non-adjusted couple’s relationship seems to boost the risk of the toddler’s psychological maladjustment. As the spillover model states, the couple’s relationship during the transition to parenthood can impact the toddler’s psychological adjustment via its impact on both parenting and co-parenting [17,18], and parents’ psychopathology [19,20]. It is speculated that these dimensions could act as underlying processes (mediators) of the couple’s relationship impact on toddlers’ psychological adjustment. Therefore, the former spillover mechanisms [38,39,40,41,42] can explain the impact of the couple’s relationship during the transition to parenthood on the toddler’s externalizing and externalizing problems.

The present study provided a comprehensive perspective of the impact of the couple’s relationship during the transition to parenthood on toddlers’ psychological adjustment. Specifically, this longitudinal study considered the importance of both mother and father reports, on the quality of the couple’s relationship and toddlers’ psychological adjustment. Some limitations should also be considered. A higher sample size could allow for testing more complex models and identifying more profiles of the couple’s relationship during the transition to parenthood. Composite scores were created for mothers’ and fathers’ reports of both the couple’s relationship and the toddler’s externalizing and internalizing problems, which may hinder the comparison of the current findings with those of other studies. All the study variables were assessed with self-report measures, which may have increased the links between the variables. Findings generalization should be taken with caution.

Findings could provide major implications for perinatal practice and research. Results could support the design of more effective couple’s parenting and coparenting interventions, supporting couples to manage their relationship during the transition to parenthood. The practical implications of the present study also indicate the requirement for policies that promote the positive dimensions of the couple’s relationship, providing working time that is family-friendly, and the development of couples’ specialized psychological support [43]. 

This study could contribute to the literature on the couple’s relationship by suggesting the impact of both positive and negative dimensions of the couple’s relationship during the transition to parenthood on toddlers’ psychological adjustment. Using the spillover model, future research can explore the underlying processes of the impact of both positive and negative dimensions of the couple’s relationship during the transition to parenthood on toddlers’ psychological adjustment [44].

## 5. Conclusions

Findings suggested the impact of both positive and negative dimensions of the couple’s relationship during the transition to parenthood on the toddler’s emotional and behavioral problems. The positive dimension of the couple’s relationship at the 1st trimester of pregnancy and the negative dimension of the couple’s relationship at 3 months postpartum were identified as the strongest predictors of the toddler’s internalizing problems, while the negative dimension of the couple’s relationship at 3 months postpartum and the positive dimension of the couple’s relationship at 30 months postpartum were the strongest predictors of the toddler’s externalizing problems. Moreover, higher levels of internalizing and externalizing problems were found in toddlers from couples with a non-adjusted couple’s relationship during the transition to parenthood. Promoting the couple’s relationship adjustment during the transition to parenthood can help to prevent toddlers’ emotional and behavioral problems.

## Figures and Tables

**Figure 1 ijerph-20-00882-f001:**
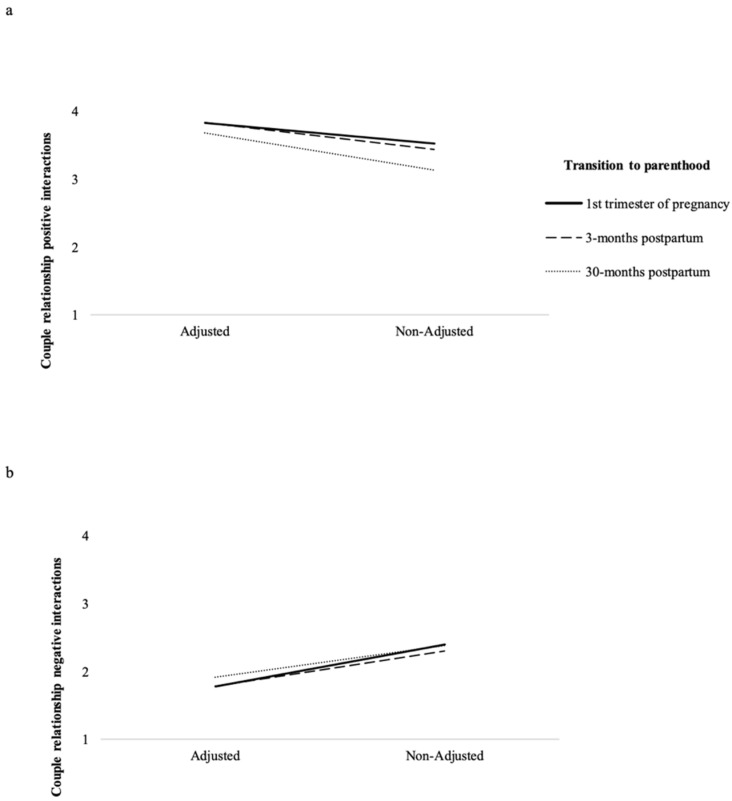
Significant interactions of time x clusters for (**a**) positive dimension of the couple’s relationship and (**b**) negative dimension of the couple’s relationship.

**Table 1 ijerph-20-00882-t001:** Participants’ socio-demographic characteristics.

		Mothers *n* = 115 (%)	Fathers *n* = 115 (%)
Age	≤19	4.4	4.3
	20–29	41.7	29.6
	30–39	51.3	55.7
	≥40	2.6	10.4
Occupational status	Employed	80.9	90.4
	Unemployed	19.1	9.6
Socio-economic level	High	23.4	23.7
	Medium	57.4	58.1
	Low	19.1	8.3
Years of education	<9	19.1	32.2
	≥9	80.9	67.8
Couple’s marital status	Single	4.3	
	Married	68.7	
	Cohabitation	27.0	

**Table 2 ijerph-20-00882-t002:** Means and standard deviations for toddler’s emotional and behavioral problems.

	Gender				Birth Order				Gestational Weeks at Birth				Birth Weight			
	Male		Female		First Born		Non-First Born		<37		≥37		<2500 g		≥2500 g	
	*M*	*SD*	*M*	*SD*	*M*	*SD*	*M*	*SD*	*M*	*SD*	*M*	*SD*	*M*	*SD*	*M*	*SD*
Internalizing	8.04	7.92	9.77	8.86	8.38	6.20	9.54	9.00	6.58	5.99	8.93	7.46	7.57	4.49	8.88	7.53
Externalizing	12.41	7.10	12.60	7.52	11.58	6.24	14.12	8.63	11.42	9.15	12.57	7.15	10.93	6.46	12.61	7.30

*Notes. M* = mean, *SD* = standard deviation. No significant effects were found (all *p*s > 0.05).

**Table 3 ijerph-20-00882-t003:** Multiple regression analysis for couple’s positive and negative dimensions of the couple’s relationship predicting toddler’s internalizing and externalizing problems.

	Internalizing				Externalizing			
	*B*	*SE B*	*β*	*p*	*B*	*SE B*	*β*	*p*
RQ positive dimension scores								
1st trimester of pregnancy	−6.50 ^2^	2.73	−0.24	0.02				
3 months postpartum								
30 months postpartum					−6.41 ^1^	2.02	−0.32	0.002
RQ negative dimension scores								
1st trimester of pregnancy								
3 months postpartum	4.71 ^1^	1.94	0.24	0.02	3.97 ^2^	1.87	0.21	0.04
30 months postpartum								
*R*	0.39				0.44			
*R* ^2^	0.15				0.20			
*R* ^2^ _adj_	0.13				0.18			
∆*R*^2^	0.05				0.04			
*F*	7.99 *				11.11 *			

*Notes. B* = non-standardized beta, *SE B* = non-standardized standard error, *β* = standardized beta, *p* = significance for *t* test, *F* = ANOVA’s value, *R* = coefficient of multiple correlation, *R*^2^
*=* R-squared*; R*^2^_adj_ = adjusted R-squared, ∆*R*^2^
*=* R-squared change. Superscript numbers refer to step entrance of predictors into the regression. Models were performed using the stepwise method. ** p* ≤ 0.001.

**Table 4 ijerph-20-00882-t004:** Means and standard deviations for clusters’ validation.

	Couple’s Relationship													
	Positive Dimension							Negative Dimension						
	1st TP		3 MPP		30 MPP		Sig. Diff.	1st TP		3 MPP		30 MPP		Sig. Diff.
	*M*	*SD*	*M*	*SD*	*M*	*SD*		*M*	*SD*	*M*	*SD*	*M*	*SD*	
Adjusted (*n* = 48)	3.83 ^a^	0.14	3.84 ^ab^	0.19	3.68 ^c^	0.34	>41.67 *	1.77 ^a^	0.34	1.78 ^ab^	0.31	1.91 ^bc^	0.36	<6.67 *
Non-Adjusted (*n* = 47)	3.52 ^a^	0.30	3.44 ^abc^	0.39	3.13 ^c^	0.64		2.40	0.26	2.30	0.29	2.38	0.36	
Time	^M^*F*(2, 92) = 15.26, *p* < 0.001, *ε*^2^_p_ = 0.25							^M^*F*(2, 92) = 2.43, *p* = 0.09, *ε*^2^_p_ = 0.05						
Clusters	^M^*F*(1, 93) = 55.00, *p* < 0.001, *ε*^2^_p_ = 0.37							^M^*F*(2, 92) = 20.76, *p* < 0.001, *ε*^2^_p_ = 0.60						
Time × Clusters	^M^*F*(2, 92) = 3.33, *p* = 0.04, *ε*^2^_p_ = 0.07							^M^*F*(2, 92) = 3.12, *p* = 0.05, *ε*^2^_p_ = 0.06						

*Notes.* 1st TP = first trimester of pregnancy, 3 MPP = 3 months postpartum, 30 MPP = 30 months postpartum. Different letters show the significant results of the mixed ANOVA interaction effects analysis after split-half procedure from adjusted and non-adjusted couples (*p*s ≤ 0.01), ^M^*F* = multivariate tests for mixed ANOVA. * = *F* and *p* for multiple comparisons (<0.01).

**Table 5 ijerph-20-00882-t005:** The impact of adjusted vs. non-adjusted couple’s relationship (clusters) on toddlers’ internalizing and externalizing problems.

	ADJ		NADJ		*MT*	*TBSE*	*PC*	*Dif.*
	*M*	*SD*	*M*	*SD*	*F*	*F*	*M* _(1−*J*)_	*_ADJ_—_NADJ_*
Internalizing	7.05	5.88	11.00	8.77	5.61 ***	6.69 *	−4.49 **	*_ADJ_* < *_NADJ_*
Externalizing	10.29	5.85	15.14	8.05		11.30 ***	−5.40 *	*_ADJ_* < *_NADJ_*

*Notes.* ADJ = adjusted relationship, NADJ = non-adjusted relationship *M* = mean, *SD* = standard deviation, *MT* = multivariate tests, *TBSE* = tests of between-subject effects, *PC* = pairwise comparisons, *M*_(1−*J*)_ = means differences. ** p ≤* 0.01, *** p* ≤ 0.005, **** p* ≤ 0.001.

## Data Availability

The data presented in this study are available on request from the corresponding author. The data are not publicly available due to ethical issues.

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
