# Peer review of "Couple’s Relationship during the Transition to Parenthood and Toddler’s Emotional and Behavioral Problems"

_ijerph, 2023, doi:10.3390/ijerph20010882_

Round 1
Reviewer 1 Report
International Journal of Environmental Research and Public Health
Introduction
“At least one study reported stable to moderate”—Is “stable” less than “moderate”? I don’t understand the range of “stable to moderate”. Please use another word or define the terms.
“…tend to decline over time and that this decline is not exclusive to couples who are transiting to parenthood”—declines in the couples’ satisfaction may happen regardless of parenthood, but the next question is, is there a steeper decline in those transitioning vs. those who only remain a couple (i.e. no kids)? If so, please add in that the decline rates are similar for couples having and not having kids.
In the paragraph “Gender is one of the major…” the authors consistently use terms like mother and father, but refer only to the couple. This paragraph would flow and be semantically more correct if terms like man and woman were used instead. For example, “Higher levels of father affection…” should be “Higher levels of male affection” as the rest of the sentence refers to the couple and has nothing to do with parenting a child.
Since an aim (Aim 2) was to analyze the impact of non-adjusted vs. adjusted couple’s relationship trajectories, could the authors add in a paragraph on what’s known regarding adjusting couple’s relationship trajectories and/or why this topic could be worth exploring. Pulling out that information should better help feed the reader into your Aim 2 and the Results of this aim.
A big part of your Introduction focused on parental gender and how there are differences between mothers and fathers regarding the couple’s relationship. However, in the data analysis section, the authors note that they’ve used a composite score derived from the maternal and paternal data. Why not explore parental gender differences (and similarities) in reporting children’s internalizing and externalizing problems? Or, if the authors want to use the composite score, then perhaps consider re-writing the Introduction and not focusing on parental gender differences.
The authors might consider shortening their paragraph on the debate about couple’s relationship quality declining (or not) over time, especially in comparison with non-parents---as this wasn’t the focus of the article.
Since the focus of this manuscript is on what the strongest predictor of toddlers’ psychological adjustment are—the authors might add in a theoretical perspective or further logical empirical evidence on why both positive and negative couple relationship quality impacts children’s internalizing and externalizing problems. The authors of course touch on this in their paragraph lines 67-77, but this could be expanded via a theory.
Methods
“The total sample (CBCL and the RQ scores) did…”The first time the child behaviour checklist is mentioned, it’s only an acronym. Please include the full name (acronym) the first time the scale is mentioned. Similarly, what are RQ scores---please report in the Methods.
“…who did not complete the data on the three assessment waves.”---it would be very helpful if the authors generally reported what data and infant age was at each of the three assessment waves. Right now, the authors briefly mention their sample size before listing socio-demographic results of their population and then mention that there were three assessment waves. So at this point, the reader is unaware of them; although a guess can be made about the third wave being at 30 months.
The authors reported that more than half of their participants were first-time parents, suggesting that a sizeable minority were multi-time parents. Please report the percent that were first-time parents. However, the authors also stated “The study exclusion criteria were… multiple gestations.” Give the context from earlier, I’m guessing this doesn’t refer to having previous children, but rather to having multiple children at the same time (e.g. twins)---please clarify the term “multiple gestations”.
“This study had a longitudinal design with three assessment waves…”—it would be great if this procedure information could be moved up, as you already refer to the three waves in the Participants section. So knowing when the three waves took place would be helpful as baseline knowledge when reading.
2.3.1 –it seems weird to state that socio-demographic information was collected, when it seems like the results of this information have already been reported. Maybe instead, move the results of the socio-demographic information to the Results section so that in the Methods you’re just reporting what data was collected (if you’re reporting the socio-demographics of the total sample of this larger project, then keep it in the Methods, but if you’re reporting on who’s in your current sample, consider moving it to the beginning of the Results).
Lines 135-136—When it comes to the toddlers’ socio-demographic data, which I supposed means infant sex and infant age, but maybe refers to additional variables (?), who completed that data collection---the mother, father or both, individually or did they do it jointly/together? I’m guessing each parent completed the survey independently since you later mentioned making a composite score in the data analysis, but this information could be stated more clearly in this section (lines 135-136).
In the CBCL, please tell the reader who completed this survey, the mother, father or both, either individually or if they completed it jointly/together. I’m guessing each parent completed the survey independently since you mentioned making a composite score in the data analysis, but this could be stated more clearly. Similarly, an alpha of 0.90 was given, but who’s data is this based on—the mothers’, fathers’ or both?
Table 1—I’m not sure what the response options were, but it seems unhelpful to group parents’ age from 20-39, since the vast majority of participants have their child during this period. If possible, perhaps go in increments of 5 years, so we can see how many 20-24 year old parents there are compared to 35-39, for example. By grouping them together, it’s possible that 90% of the parents are all 30-34, but under the heading of 20-39, which is why it be nice to break this down a bit more.
In the aim, the authors write that their goal is to “to identify which moment of the positive and negative dimensions of the couple’s relationship during the transition to parenthood is the stronger predictor of the toddler’s psychological adjustment.” When I read the word “moment” I thought that this meant if affection was a more important factor than closeness joint interests OR if criticism is more important than anxiety. However, in the Abstract, the authors seem to relate “moment” to the time period for the child (e.g. 1st pregnancy trimester or 3 months postpartum). Therefore, the authors may want to clarify/define the word “moment” in their aim. Moreover, beyond the abstract, the term “strongest predictor” is never used in the Results or Discussion section, even though that was the stated aim. It might therefore be helpful to the reader if the authors updated the Results and/or Discussion to highlight which moments were the strongest predictors using that term.
Limitations
The authors have chosen to combine the maternal and paternal data for their analyses. However, in the current draft, the Introduction makes it clear that mothers and fathers don’t rate their relationship or children’s internalizing/externalizing problems the same. If no analyses are comparing mothers’ and fathers’ ratings of their child (e.g. only composite scores are used), then the authors might consider mentioning any limitations that composite scores might yield. While there may be some recommendations to this, the Introduction is showing that fathers and mothers rank differently, and so if another author wants to compare their findings to yours, they may not be so comparable unless they also could make a composite score (e.g. not only have mother data or only have father data).
Grammar
There are some grammatical issues throughout the manuscript that require a second reading. For example, the authors write “The 36 dyad copes with this developmental transition a…”—I can’t figure out this sentences’ meaning and so can’t correct the “a” at the end of the phrase.
Another example: “on features of the couples who transit to parenthood”—“transit” should be “transition”.
“Literature has been reporting”—this could be “Literature reports”
These are just examples though, rather than a thorough grammatical review.
Author Response
First, we would like to thank the reviewer for their comments on the manuscript, which we have addressed thoroughly. The changes performed in the manuscript were highlighted using the track changes mode. We included the response to each of your comments in the text below.
Reviewer’s comment 1
Introduction
“At least one study reported stable to moderate”—Is “stable” less than “moderate”? I don’t understand the range of “stable to moderate”. Please use another word or define the terms.
Response to reviewer’s comment 1
According to your comment number 6, we revised this paragraph (please see page 2, lines 51-62).
Reviewer’s comment 2
“…tend to decline over time and that this decline is not exclusive to couples who are transiting to parenthood”—declines in the couples’ satisfaction may happen regardless of parenthood, but the next question is, is there a steeper decline in those transitioning vs. those who only remain a couple (i.e. no kids)? If so, please add in that the decline rates are similar for couples having and not having kids.
Response reviewer’s comment 2
According to your comment number 6, we revised this paragraph (please see page 2, lines 51-62).
Reviewer’s comment 3
In the paragraph “Gender is one of the major…” the authors consistently use terms like mother and father, but refer only to the couple. This paragraph would flow and be semantically more correct if terms like man and woman were used instead. For example, “Higher levels of father affection…” should be “Higher levels of male affection” as the rest of the sentence refers to the couple and has nothing to do with parenting a child.
Response to reviewer’s comment 3
We totally agree with your suggestion! However, considering your comment number 5 (that we also agree with), we decided to remove this paragraph as this is not the focus of the study (please see page 2, lines 68-75).
Reviewer’s comment 4
Since an aim (Aim 2) was to analyze the impact of non-adjusted vs. adjusted couple’s relationship trajectories, could the authors add in a paragraph on what’s known regarding adjusting couple’s relationship trajectories and/or why this topic could be worth exploring. Pulling out that information should better help feed the reader into your Aim 2 and the Results of this aim.
Response to reviewer’s comment 4
As you suggested, we added some sentences about the knowledge and importance of exploring profiles of couple’s relationship trajectories during the transition to parenthood (please see page 2, lines 62-67) and its impact on children’s psychological adjustment (please see page 3, lines 102-105).
Reviewer’s comment 5
A big part of your Introduction focused on parental gender and how there are differences between mothers and fathers regarding the couple’s relationship. However, in the data analysis section, the authors note that they’ve used a composite score derived from the maternal and paternal data. Why not explore parental gender differences (and similarities) in reporting children’s internalizing and externalizing problems? Or, if the authors want to use the composite score, then perhaps consider re-writing the Introduction and not focusing on parental gender differences.
Response to reviewer’s comment 5
We totally agree with you! Thank you for this relevant comment. In fact, when we started to design this manuscript, we also aimed to explore possible gender differences. However, when conducting power analyses, we realized that our sample size is low to also include in the regression models gender and their interaction with couple’s relationship and children’s internalizing and externalizing problems. This is one of the main reasons why we decided to move forward with the composite scores. As you suggested, we re-write the introduction avoiding the focus on parental gender differences.
Reviewer’s comment 6
The authors might consider shortening their paragraph on the debate about couple’s relationship quality declining (or not) over time, especially in comparison with non-parents---as this wasn’t the focus of the article.
Response to reviewer’s comment 6
As you suggested, we shortened this paragraph (please see page 2, lines 51-62).
Reviewer’s comment 7
Since the focus of this manuscript is on what the strongest predictor of toddlers’ psychological adjustment are—the authors might add in a theoretical perspective or further logical empirical evidence on why both positive and negative couple relationship quality impacts children’s internalizing and externalizing problems. The authors of course touch on this in their paragraph lines 67-77, but this could be expanded via a theory.
Response to reviewer’s comment 7
As you suggested, we expanded this issue via a theory (please see 2, lines 76-78).
Reviewer’s comment 8
Methods
“The total sample (CBCL and the RQ scores) did…”The first time the child behaviour checklist is mentioned, it’s only an acronym. Please include the full name (acronym) the first time the scale is mentioned. Similarly, what are RQ scores---please report in the Methods.
Response to reviewer’s comment 8
Considering your comment number 12, we moved this subsection to the results section. Thus, we have maintained the acronym.
Reviewer’s comment 9
“…who did not complete the data on the three assessment waves.”---it would be very helpful if the authors generally reported what data and infant age was at each of the three assessment waves. Right now, the authors briefly mention their sample size before listing socio-demographic results of their population and then mention that there were three assessment waves. So at this point, the reader is unaware of them; although a guess can be made about the third wave being at 30 months.
Response to reviewer’s comment 9
We totally agree with you! We think it is clearer since we moved this section to the results section.
Reviewer’s comment 10
The authors reported that more than half of their participants were first-time parents, suggesting that a sizeable minority were multi-time parents. Please report the percent that were first-time parents. However, the authors also stated “The study exclusion criteria were… multiple gestations.” Give the context from earlier, I’m guessing this doesn’t refer to having previous children, but rather to having multiple children at the same time (e.g. twins)---please clarify the term “multiple gestations”.
Response to reviewer’s comment 10
As suggested, we added the percentage of first-time parents (please see page 5, line 228) and clarified the term “multiple gestations” (please see page 4, line 142).
Reviewer’s comment 11
“This study had a longitudinal design with three assessment waves…”—it would be great if this procedure information could be moved up, as you already refer to the three waves in the Participants section. So knowing when the three waves took place would be helpful as baseline knowledge when reading.
Response to reviewer’s comment 11
According to your suggestions, the procedure information is now in the beginning of the methods section.
Reviewer’s comment 12
2.3.1 –it seems weird to state that socio-demographic information was collected, when it seems like the results of this information have already been reported. Maybe instead, move the results of the socio-demographic information to the Results section so that in the Methods you’re just reporting what data was collected (if you’re reporting the socio-demographics of the total sample of this larger project, then keep it in the Methods, but if you’re reporting on who’s in your current sample, consider moving it to the beginning of the Results).
Response to reviewer’s comment 12
According to your suggestion, we moved this information to the results section.
Reviewer’s comment 13
Lines 135-136—When it comes to the toddlers’ socio-demographic data, which I supposed means infant sex and infant age, but maybe refers to additional variables (?), who completed that data collection---the mother, father or both, individually or did they do it jointly/together? I’m guessing each parent completed the survey independently since you later mentioned making a composite score in the data analysis, but this information could be stated more clearly in this section (lines 135-136).
Response to reviewer’s comment 13
As you suggested, we clearly state this information in the measures section (please see page 4, lines 154-157). This information is also stated in the procedures section (please see page 4, lines 148-151).
Reviewer’s comment 14
In the CBCL, please tell the reader who completed this survey, the mother, father or both, either individually or if they completed it jointly/together. I’m guessing each parent completed the survey independently since you mentioned making a composite score in the data analysis, but this could be stated more clearly. Similarly, an alpha of 0.90 was given, but who’s data is this based on—the mothers’, fathers’ or both?
Response to reviewer’s comment 14
As you suggested, we stated this more clearly in the measures section (please see page 4, lines 172-173). This information is also stated in the procedures section (please see page 4, lines 148-151). We also clarified the alpha values (please see page 5, lines 183-184).
Reviewer’s comment 15
Table 1—I’m not sure what the response options were, but it seems unhelpful to group parents’ age from 20-39, since the vast majority of participants have their child during this period. If possible, perhaps go in increments of 5 years, so we can see how many 20-24 year old parents there are compared to 35-39, for example. By grouping them together, it’s possible that 90% of the parents are all 30-34, but under the heading of 20-39, which is why it be nice to break this down a bit more.
Response to reviewer’s comment 15
As you suggested, we broke down the age groups (please see Table 1).
Reviewer’s comment 16
In the aim, the authors write that their goal is to “to identify which moment of the positive and negative dimensions of the couple’s relationship during the transition to parenthood is the stronger predictor of the toddler’s psychological adjustment.” When I read the word “moment” I thought that this meant if affection was a more important factor than closeness joint interests OR if criticism is more important than anxiety. However, in the Abstract, the authors seem to relate “moment” to the time period for the child (e.g. 1st pregnancy trimester or 3 months postpartum). Therefore, the authors may want to clarify/define the word “moment” in their aim. Moreover, beyond the abstract, the term “strongest predictor” is never used in the Results or Discussion section, even though that was the stated aim. It might therefore be helpful to the reader if the authors updated the Results and/or Discussion to highlight which moments were the strongest predictors using that term.
Response to reviewer’s comment 16
Trying to address your suggestion, we replaced “moment” by “period” in the study aims and defined it (please see page 3, lines 109-110). We also revised the results and discussion sections using the term strongest predictor.
Reviewer’s comment 17
Limitations
The authors have chosen to combine the maternal and paternal data for their analyses. However, in the current draft, the Introduction makes it clear that mothers and fathers don’t rate their relationship or children’s internalizing/externalizing problems the same. If no analyses are comparing mothers’ and fathers’ ratings of their child (e.g. only composite scores are used), then the authors might consider mentioning any limitations that composite scores might yield. While there may be some recommendations to this, the Introduction is showing that fathers and mothers rank differently, and so if another author wants to compare their findings to yours, they may not be so comparable unless they also could make a composite score (e.g. not only have mother data or only have father data).
Response to reviewer’s comment 17
Thank you so much for your careful review. As you suggested, we included this issue in the limitations section (please see page 11, lines 397-399).
Reviewer’s comment 18
Grammar
There are some grammatical issues throughout the manuscript that require a second reading. For example, the authors write “The 36 dyad copes with this developmental transition a…”—I can’t figure out this sentences’ meaning and so can’t correct the “a” at the end of the phrase.
Response to reviewer’s comment 18
We revised the sentence to make it clear (please see page 1).
Reviewer’s comment 19
Another example: “on features of the couples who transit to parenthood”—“transit” should be “transition”.
Response to reviewer’s comment 19
Amended, as suggested (please see page 2, line 46).
Reviewer’s comment 20
“Literature has been reporting”—this could be “Literature reports”
Response to reviewer’s comment 20
Amended, as suggested (please see page 2, line 47).
Reviewer’s comment 21
These are just examples though, rather than a thorough grammatical review.
Response to reviewer’s comment 21
We revised the manuscript throughout.
Reviewer 2 Report
Thank you very much for having shared to me your interesting paper.
The research submitted to the journal investigates the transition to parenthood of a couple connected with health of the child.
The title, "Couple's relationship during the transition to parenthood and toddler emotional and behavioral problems", perfectly clarifies the topic covered throughout the article.
The keywords, line 30 do not help to index the article correctly. I suggest inserting “emotional and behavioral problems”.
Firstly, the abstract is well written, it describes the steps of the article in a clear manner. I would just add a few notes on methods, such as the questionnaires used and the main statistics.
Likewise, the introduction describes the background of the work, explaining the aim of the current research, summarizing the specific objectives as well (line 93-98).
On the other hand, the description of the research is slightly very simple, the authors could expand the description of this section (sample, research methods and data analyze).
Likewise, the two questionnaires used, CBCL (Child Behavior Checklist) and Relationship Questionnaire (RQ), are simply listed. As a consequence, a broader description of the selected tools and how they have been used in the literature for the same purpose (including psychometric analyzes carried out) is recommended. “2.3.2. Couple's relationship quality" from line 137; “2.3.3. Emotional and behavioral problems” from line 147.
Finally, in the discussions, there is a no an appropriate connection with the research described in the introduction, in order to explain the differences in the current results. Also, the methodological limits of the research and the strengths should be better described, suggesting new research studies.
The conclusions section, from line 347, is too short. I recommend to authors to rewrite it after the aforementioned revisions.
Good luck for your studies
Author Response
First, we would like to thank the reviewer for their comments on the manuscript, which we have addressed thoroughly. The changes performed in the manuscript were highlighted using the track changes mode. We included the response to each of your comments in the text below.
Reviewer’s comment 1
Thank you very much for having shared to me your interesting paper.
The research submitted to the journal investigates the transition to parenthood of a couple connected with health of the child.
Response to reviewer’s comment 1
Thank you so much!
Reviewer’s comment 2
The title, "Couple's relationship during the transition to parenthood and toddler emotional and behavioral problems", perfectly clarifies the topic covered throughout the article.
Response to reviewer’s comment 2
Thank you for your careful reading.
Reviewer’s comment 3
The keywords, line 30 do not help to index the article correctly. I suggest inserting “emotional and behavioral problems”.
Response to reviewer’s comment 3
We have inserted it in the keywords (please see page 1, line 31).
Reviewer’s comment 4
Firstly, the abstract is well written, it describes the steps of the article in a clear manner. I would just add a few notes on methods, such as the questionnaires used and the main statistics.
Response to reviewer’s comment 4
As you suggested, we described the questionnaires and the main statistics.
Reviewer’s comment 5
Likewise, the introduction describes the background of the work, explaining the aim of the current research, summarizing the specific objectives as well (line 93-98).
Response to reviewer’s comment 5
Thank you so much!
Reviewer’s comment 6
On the other hand, the description of the research is slightly very simple, the authors could expand the description of this section (sample, research methods and data analyze).
Response to reviewer’s comment 6
We revised these sections to expand their description.
Reviewer’s comment 7
Likewise, the two questionnaires used, CBCL (Child Behavior Checklist) and Relationship Questionnaire (RQ), are simply listed. As a consequence, a broader description of the selected tools and how they have been used in the literature for the same purpose (including psychometric analyzes carried out) is recommended. “2.3.2. Couple's relationship quality" from line 137; “2.3.3. Emotional and behavioral problems” from line 147.
Response to reviewer’s comment 7
As you suggested, we improved the description of the selected tools (please see page 4, lines 159-185).
Reviewer’s comment 8
Finally, in the discussions, there is a no an appropriate connection with the research described in the introduction, in order to explain the differences in the current results. Also, the methodological limits of the research and the strengths should be better described, suggesting new research studies.
Response to reviewer’s comment 8
We revised the discussion section to address your suggestions.
Reviewer’s comment 9
The conclusions section, from line 347, is too short. I recommend to authors to rewrite it after the aforementioned revisions.
Response to reviewer’s comment 9
As you suggested, we revised the conclusion.
Reviewer’s comment 10
Good luck for your studies
Response to reviewer’s comment 10
Thank you so much!
Round 2
Reviewer 1 Report
The authors have made substantial improvements to the manuscript, and have adjusted the manuscript according to the reviewers' (my) comments or defended their position in a coherent way.
For example, I'd agree that their sample size would make it difficult to compare and contact mothers vs. fathers.
I think the way the authors handled their analysis and wrote their interpretation makes logical sense and I have no new comments to add.
Well done.